# A Novel Digital Technique to Analyze the Wear of CM-Wire NiTi Alloy Endodontic Reciprocating Files: An In Vitro Study

**DOI:** 10.3390/ijerph19063203

**Published:** 2022-03-09

**Authors:** Vicente Faus-Matoses, Vicente Faus-Llácer, Álvaro Aldeguer Muñoz, Jorge Alonso Pérez-Barquero, Ignacio Faus-Matoses, Celia Ruiz-Sánchez, Álvaro Zubizarreta-Macho

**Affiliations:** 1Department of Stomatology, Faculty of Medicine and Dentistry, University of Valencia, 46010 Valencia, Spain; vicente.faus@uv.es (V.F.-M.); fausvj@uv.es (V.F.-L.); alalmu@alumni.uv.es (Á.A.M.); jorgealonso86@gmail.com (J.A.P.-B.); ceruizsan@gmail.com (C.R.-S.); 2Department of Endodontics, Faculty of Health Sciences, Alfonso X El Sabio University, 28691 Madrid, Spain; amacho@uax.es; 3Department of Surgery, Faculty of Medicine and Dentistry, University of Salamanca, 37008 Salamanca, Spain

**Keywords:** endodontics, cyclic fatigue, wear, NiTi, reciprocating movement, micro-computed tomography

## Abstract

The present study seeks to describe a novel digital measurement technique for analyzing the wear volume of controlled memory (CM)-wire NiTi alloy endodontic reciprocating files after clinical use. Material and Methods: Ten CM-wire NiTi endodontic reciprocating files were randomly used in ten first upper molar teeth within four root canals. The CM-wire NiTi alloy endodontic reciprocating files were subjected to preoperative and postoperative micro-computed tomography (micro-CT) scans to obtain accurate digital imaging and communication on medicine (DICOM) digital files, which were segmented using intensity-based thresholding and an exclusive OR (XOR) logical operation (Boolean algebra logical operator) to obtain a mask of the location to localize and quantify the wear volume of the CM-wire NiTi alloy endodontic reciprocating files. Gage repeatability and reproducibility statistical analysis was applied to assess the reproducibility and repeatability of this measurement technique. Results: The analysis showed a repeatability and reproducibility of 0.00% for the digital measurement technique used to analyze the wear volume of CM-wire NiTi alloy endodontic reciprocating files after clinical use. Wear was mostly identified at the tip and cutting edges of the CM-wire NiTi alloy endodontic reciprocating files. Conclusions: This novel digital measurement technique is a repeatable, reproducible, and accurate method of quantifying the wear volume of CM-wire NiTi alloy endodontic reciprocating files after clinical use.

## 1. Introduction

The removal of bacteria from the root canal system is one of the most significant factors influencing the prognosis of root canal treatment [1]; as a result, techniques for cleaning and shaping procedures are continuously being developed to help reduce bacterial loads within the root canal system with a view to preventing endodontic failure caused by persistent or secondary dental infections [2]. Additionally, novel trends are being developed to promote dentin–pulp formation using polymer-based instructive scaffolds [3,4].

There is also the ongoing development of endodontic rotary files, regarding both the chemical composition of metal alloys and their geometrical design, which improves the mechanical resistance of endodontic rotary files to bending and torsional stress [5]. Other characteristics of the geometrical design of endodontic rotary files that reportedly influence instrument performance include the cross section [6]; the helix angle, flute length, and pitch [7]; and the taper and apical diameter [4]. More specifically, a larger apical diameter and taper are correlated with lower cyclic fatigue resistance in nickel–titanium (NiTi) endodontic rotary files, with the files becoming more liable to fracture unexpectedly [3]. NiTi endodontic rotary files fracture at a rate of between 0.09% and 5% [8,9], with both flexural (bending) and torsional failure having been identified as the primary factors leading to endodontic rotary file fractures [10,11]. When endodontic files become blocked during movement, this can result in torsional failure [8], and if they rotate within a curved root canal, this can lead to cyclic bending fatigue [12]. The root canal shaping process can potentially apply simultaneous cumulative torsional and bending stress, potentially leading to wear and tear, affecting the structure of NiTi endodontic rotary files and even leading to their fracture. This negatively affects the prognosis of root canal treatment because the fractured piece impedes access to the apex, making it difficult to disinfect the root canal system [13]. Shen et al. found that NiTi endodontic rotary instruments most often fracture in the apical 3–5 mm [14], which indicates that the last millimeters of the files are subjected to more flexural and torsional failure during shaping. Yared proposed a novel root canal preparation technique that employs a non-specific clockwise-cutting rotary file in a reciprocating movement [15] based on the balanced force technique [16]. NiTi endodontic reciprocating files have subsequently been developed with a view to improving and simplifying the shaping of the root canal system. Since these systems were first developed, these single-file NiTi endodontic reciprocating systems have gained popularity in the dental industry. In addition, manufacturers have also developed several thermally treated NiTi alloys with controlled memory wire that optimizes their microstructure and transformation behavior. This, in turn, has more of an impact on the mechanical behavior of NiTi endodontic reciprocating files [17]. However, torsional and bending stress can damage the structure of NiTi endodontic reciprocating files during the shaping process, leading to wear and deformation [18], negatively impacting cutting efficiency and increasing the risk of unexpected fracture [19,20]. Different measurement techniques have been used to assess the wear and deformation of NiTi endodontic files, including scanning electron microscopy (SEM) [13] and stereomicroscopy [18]. However, these measurement techniques cannot accurately analyze the volume of NiTi endodontic rotary files lost after clinical use.

The present study seeks to describe a novel digital measurement technique for analyzing the wear volume of controlled memory (CM)-wire NiTi alloy endodontic reciprocating files after clinical use, with a null hypothesis (H_0_) that there is no difference between the morphometric measurement protocols used when it comes to accurately measuring the wear level of CM-wire NiTi alloy endodontic reciprocating files after use.

## 2. Materials and Methods

### 2.1. Study Design

Ten CM-wire NiTi alloy endodontic reciprocating files (R40, Reciproc; VDW, Munich, Germany) were randomly (Epidat 4.1, Galicia, Spain) used in ten first upper molar teeth in four root canals extracted for periodontal reasons using Schneider’s method. All of the selected teeth had curvature angles of ≤20°, [21], a root length of greater than 16 mm, mature roots, and no prior incidence of root canal treatment, root resorption, or calcium metamorphosis. The study was carried out between July 2020 and October 2021 at the Department of Stomatology of the University of Valencia (Valencia, Spain). A randomized, controlled experimental trial was carried out in accordance with the statement of the German Ethics Committee on using organic tissues for medical research. The study was approved by the ethics committee of the University of Valencia (Process No. 12151). All patients provided their informed consented for their teeth to be transferred for use in the study.

### 2.2. Experimental Procedure

Tooth crowns were removed under the cement–enamel junction to achieve a standardized root length of 16 mm for all teeth. They were then scanned via digital radiography in both the buccolingual and mesiodistal directions to evaluate the root canal system anatomy. The root canal systems were performed using sterile 40.06 CM-wire NiTi alloy endodontic reciprocating files (Reciproc; VDW, Munich, Germany). The files had been analyzed under magnification (SZR-10, Optika, Bergamo, Italy) to rule out any surface defects from the manufacturing process. Next, the working length was determined by inserting a size 8 K file (Dentsply Maillefer, Baillagues, Switzerland) into the root canal system until it became visible at the apical foramen under the operative microscope at a magnification of 10 × (Zeiss Dental Microscope, Oberkochen, Germany). The root canal systems were carried out with a 6:1 reduction handpiece (Reciproc; VDW, Munich, Germany) using the crown-down technique with a torque-controlled motor with reciprocant motion (Reciproc; VDW, Munich, Germany) for a duration of 73.1 s, according to the study performed by Bürklein et al. [22]. The same clinician performed all root canal treatments.

### 2.3. Micro-CT Scanning Procedures and Evaluation

The files were subjected to preoperative and postoperative micro-computed tomography (micro-CT) scans (Skyscan 1176, Bruker-MicroCT, Kontich) with the following exposure parameters: 56.0–58.0 microamperes, 160.0 kilovolt peak, 720 projections, 500.0 msec, 4 frames, a 3 µm resolution, a tungsten target between 0.25 and 0.375 mm, and a pixel size of 0.127 µm. These scans provided DICOM digital files of the endodontic files. The first micro-CT scan (Micro-CAT II, Siemens Preclinical Solutions, Knoxville, TN, USA) was taken prior to the root canal treatment (DICOM1) (Figure 1A), and the second micro-CT scan was taken after the root canal treatments had been completed (DICOM2) (Figure 1B).

### 2.4. Alignment Procedure

Once the preoperative (DICOM1) and postoperative (DICOM2) digital files from the micro-CT scans of the endodontic files had been aligned into a common coordinate space by image registration using an in-house MeVisLab network (MeVis Medical Solutions AG), the digital files were further processed using Fiji/ImageJ, an open-source, Java-based image processing software product [23]. The preoperative (DICOM1) and postoperative (DICOM2) digital files were subsequently segmented via intensity-based thresholding. An exclusive OR (XOR) logical operation (Boolean algebra logical operator) was then applied to obtain a mask of the wear location of the endodontic files (Figure 2).

### 2.5. Measurement Procedure

After the alignment procedure, the volume of the resulting masks (mm^3^) of the preoperative (DICOM1) and postoperative (DICOM2) digital files from the micro-CT scans of the CM-wire NiTi alloy endodontic reciprocating files was assessed, with the active part segmented to quantify the level of wear caused to the files. Finally, 3D images were rendered using Amira 3D software (Termo Scientific, Waltham, MA, USA) for preclinical analysis. The locations of wear were analyzed between the DICOM1 and DICOM2 digital files (Figure 3A) at 1 mm (Figure 3B), 2 mm (Figure 3C), 3 mm (Figure 3D), 4 mm (Figure 3E), 5 mm (Figure 3F), and 6 mm (Figure 3G) from the tip of the CM-wire NiTi alloy endodontic reciprocating files. All the plugins were developed by the imaging platform at the Center for Applied Medical Research (Navarra, Spain).

In addition, the aligned preoperative (DICOM1) and postoperative (DICOM2) digital files from the micro-CT scans of the CM-wire NiTi alloy endodontic reciprocating files (Figure 4B) were isolated (Figure 4B) and measured in volume (Figure 4C) to analyze the wear of the CM-wire NiTi alloy endodontic reciprocating files after use.

### 2.6. Confirmation of Repeatability and Reproducibility

To confirm the repeatability of this novel digital measurement technique, a single operator (operator A) calculated the aforementioned measurements six times. A second operator (operator B) calculated the measurements six times to validate the reproducibility of this novel digital measurement technique.

### 2.7. Statistical Tests

Statistical analysis was conducted using SAS 9.4 (SAS Institute Inc., Cary, NC, USA) to evaluate the measured valuables. Descriptive statistics are expressed as the mean and standard deviation (SD) for quantitative variables. Gage repeatability and reproducibility statistical analysis was carried out to assess the reproducibility and repeatability of this measurement technique.

## 3. Results

Table 1 shows the mean and SD values for the wear volume (mm^3^) of the CM-wire NiTi alloy endodontic reciprocating files between the operators.

Additionally, wear was mostly registered at the tip and cutting edges of the CM-wire NiTi alloy endodontic reciprocating files.

The Gage repeatability and reproducibility statistical analysis of the wear volume observed in the CM-wire NiTi alloy endodontic reciprocating files after use found that the variability attributable to the digital technique for measuring wear volume had a repeatability value of 0.00% of the total variability of the samples. As repeatability values must be below 1% in order to constitute high repeatability, the morphometric measurement technique demonstrates a high repeatability rate for volume measurement. Moreover, the correlation coefficient among operators was 1. In addition, the Gage repeatability and reproducibility statistical analysis of the wear volume of the files after use revealed that the variability attributable to the measurement technique performed by the two operators had a reproducibility value of 0.00% of the total variability of the samples. Reproducibility values must be below 1% to demonstrate high reproducibility; therefore, it can be concluded that this morphometric measurement technique has a high level of reproducibility when evaluating the wear volume in CM-wire NiTi alloy endodontic reciprocating files after use (see Figure 5 and Figure 6).

## 4. Discussion

The results of the present study reject the null hypothesis (H_0_) that there is no difference between morphometric measurement protocols in the accuracy of measuring the volume wear of CM-wire NiTi alloy endodontic reciprocating files after use.

The digital technique presented herein provides a repeatable, reproducible, and accurate method for quantifying the volume wear of endodontic reciprocating files after use in clinical settings.

The fatigue resistance of endodontic rotary or reciprocating files may be influenced by the metallurgical characteristics of the NiTi alloy, which affect the physical properties of the files [24,25]. Additionally, the kinematics of the NiTi alloy endodontic rotary files may also influence the cyclic fatigue resistance to rotary or reciprocating movement, increasing the resistance to cyclic fatigue of the endodontic files [26]. Furthermore, pecking motion frequency has been reported to affect the resistance to cyclic fatigue of NiTi alloy endodontic rotary files [27].

The geometrical parameters of the NiTi alloy endodontic files may also influence the level of resistance to cyclic fatigue of NiTi alloy endodontic rotary and reciprocating files. Sekar et al. [6] and Zhang et al. [28] analyzed the effect of cross-section design on resistance to cyclic fatigue, finding that apical diameter and taper were linked to a higher risk of fracture.

However, speed does not affect the resistance to cyclic fatigue of NiTi alloy endodontic rotary files [29,30], unless the endodontic rotary file is already close to reaching the critical number of rotations [31].

Surface wear located at the cutting edge of NiTi alloy endodontic rotary files may also influence the effectiveness of endodontic instruments during use [14]. Spicciareli et al. reported morphological alterations and reduced cutting efficiency of CM-wire NiTi alloy endodontic reciprocating files after repeated use by analyzing seven deteriorating variables assessed using micrographs taken before and after use. Fatigue cracks and metal strips/metal flash were observed in the gradual wear of the cutting edge, which could negatively affect the cutting efficiency of the NiTi alloy endodontic reciprocating files [32]. In addition, Zubizarreta-Macho et al. pointed to the influence of the number of uses on resistance to cyclic fatigue of NiTi alloy endodontic rotary files [33], and Vieira et al. observed the appearance of transversal crack defects in the surface of NiTi alloy endodontic rotary files under flexural stress, as well as longitudinal crack defects in the surface of files after use. They reported that the cumulative effect of fatigue on NiTi alloy endodontic rotary files more significantly influences flexural fatigue conduct in comparison with torsional resistance [34].

Different protocols have been used to measure wear volume in NiTi alloy endodontic rotary files. Peters et al. used a high-resolution tomography scan to observe changes in extracted teeth before and after the use of NiTi K-Files, Lightspeed, ProFile.04 Instruments, and GT-Rotaries; however, the pre- and post-operative images were not aligned to evaluate the localization of wear [34]. Aracena et al. analyzed the deterioration of Wave One CM-wire NiTi alloy reciprocating files under a microscope after cutting them into three cross-sections, observing differences in irregularities and rake angles between the unused and used instruments [35]. In the present study, the NiTi alloy endodontic rotary files were virtually sectioned into six cross-sections. Inan et al. assessed the deformation and incidence of fracture in Mtwo NiTi alloy endodontic rotary files after clinical use by measuring the total length of the Mtwo NiTi alloy endodontic rotary files with a digital caliper; however, this provides a more subjective measurement procedure [18].

Tripi et al. assessed the effect of NiTi alloy endodontic rotary instruments using SEM analysis, identifying small holes, cracks, and detritus that may increase the potential risk of failure and reduce the cutting efficiency of NiTi alloy endodontic rotary instruments [3]. In addition, Inan et al. analyzed structural defects in NiTi alloy endodontic rotary instruments using stereomicroscopy and reported cracks, untwisting, loops, or deflections.

In the past, micro-CT analysis has been used to analyze the effects of root canal preparations on canal volume and tooth surface. Peters et al. studied the effects of four different shaping systems on the removal of dentine using 3D replicas of root canal systems of upper molar teeth [36]. Connert et al. used a micro-CT scan to assess the accuracy of electronic apex locators in determining the levels of apical constriction during root canal treatment [37]. Zuolo et al. also used a micro-CT scan to measure the frequency of microcracks caused by root canal preparation with TRUShape and Self-Adjusting Files [38]. In addition, the micro-CT measurement technique has been used to analyze the amount of root canal filling material remaining inside the root canal system after root canal retreatment [39], and Perez et al. evaluated the relationship between the depth of insertion of the irrigation needle and the removal of hard tissue detritus [39]. Micro-CT scans have also been used to assess anatomical changes in the root canal system after preparation, with findings indicating that NiTi alloy endodontic rotary instruments can cause morphological alterations in the root canal system anatomy directly linked to the geometrical design and metallurgical alloy of the file. Root canal anatomy also has a large impact on the way that an endodontic instrument works. However, micro-CT scans had not previously been used to analyze changes in the surface of NiTi alloy endodontic rotary files after use.

In the present study, we used CM-wire NiTi alloy endodontic reciprocating files to prevent flexion and torsion of the files during use, which prevents the alignment of DICOM files for the purpose of analyzing the wear location. One limitation observed in the present study is that NiTi alloy endodontic files with smaller diameters, as well as files with metallurgical composition or thermal treatments that allow more flexibility, may compromise the alignment procedure.

## 5. Conclusions

The results of the present study indicate that this novel digital measurement technique is a reproducible, repeatable, and accurate method for quantifying the volume wear of CM-wire NiTi alloy endodontic reciprocating files after clinical use.

## Figures and Tables

**Figure 1 ijerph-19-03203-f001:**
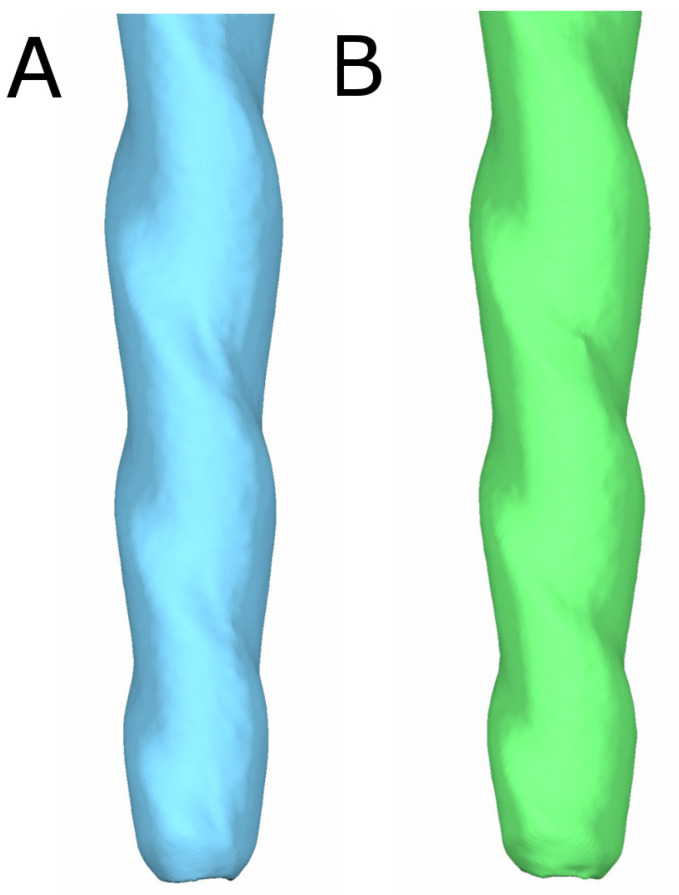
Three-dimensional reconstruction of the (**A**) preoperative and (**B**) postoperative micro-CT scans of a CM-wire NiTi alloy endodontic reciprocating file.

**Figure 2 ijerph-19-03203-f002:**
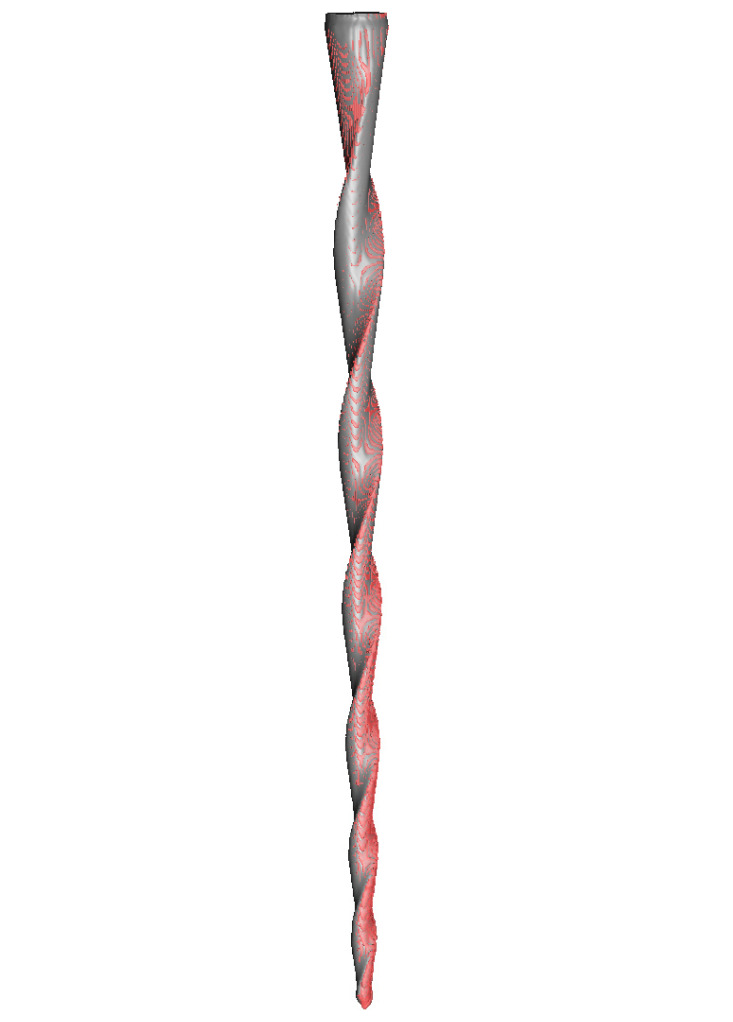
The procedure for alignment between the preoperative (red) and postoperative (grey) digital files and identification of the localization of the CM-wire NiTi alloy endodontic reciprocating files.

**Figure 3 ijerph-19-03203-f003:**
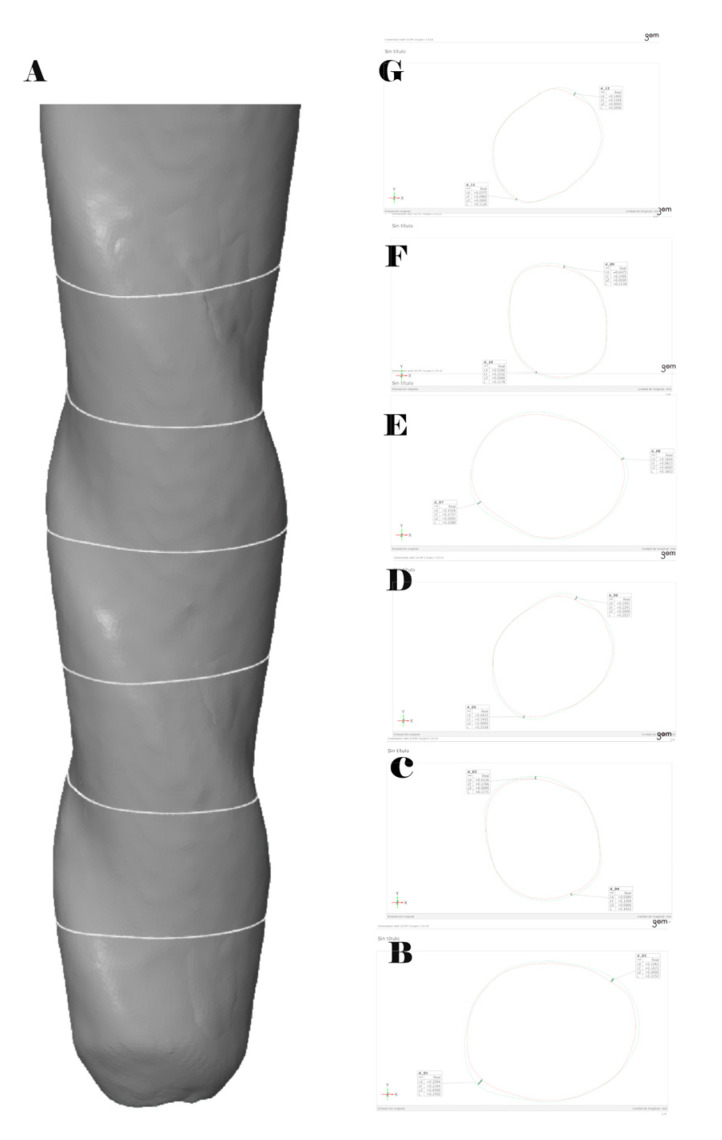
(**A**) Three-dimensional reconstruction of the aligned preoperative and postoperative digital files of the CM-wire NiTi alloy endodontic reciprocating files and cross sections at (**B**) 1 mm, (**C**) 2 mm, (**D**) 3 mm, (**E**) 4 mm, (**F**) 5 mm, and (**G**) 6 mm from the tip of the files.

**Figure 4 ijerph-19-03203-f004:**
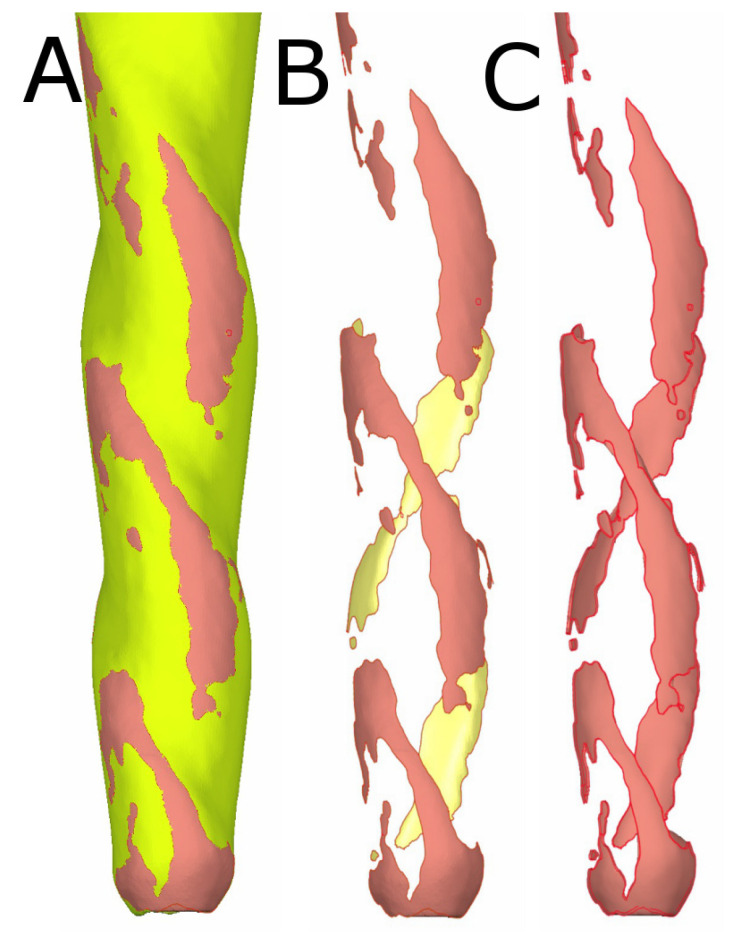
(**A**) Three-dimensional reconstruction of the aligned preoperative (yellow) and postoperative (red) digital files to isolate and measure the (**B**) wear area and (**C**) wear volume.

**Figure 5 ijerph-19-03203-f005:**
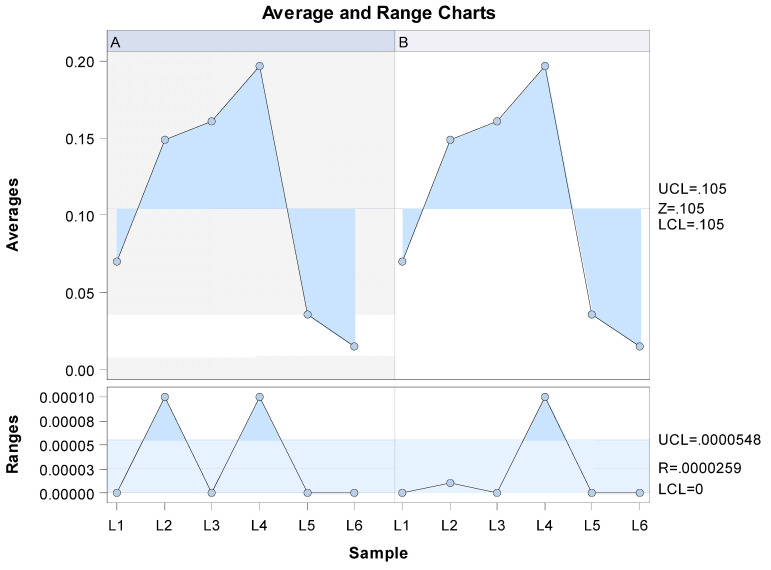
Mean values of the six measurements of the volume wear of CM-wire NiTi alloy endodontic reciprocating files after use, performed by operators A and B (average), and the differences between the six measurements of the volume wear of CM-wire NiTi alloy endodontic reciprocating files after use (range).

**Figure 6 ijerph-19-03203-f006:**
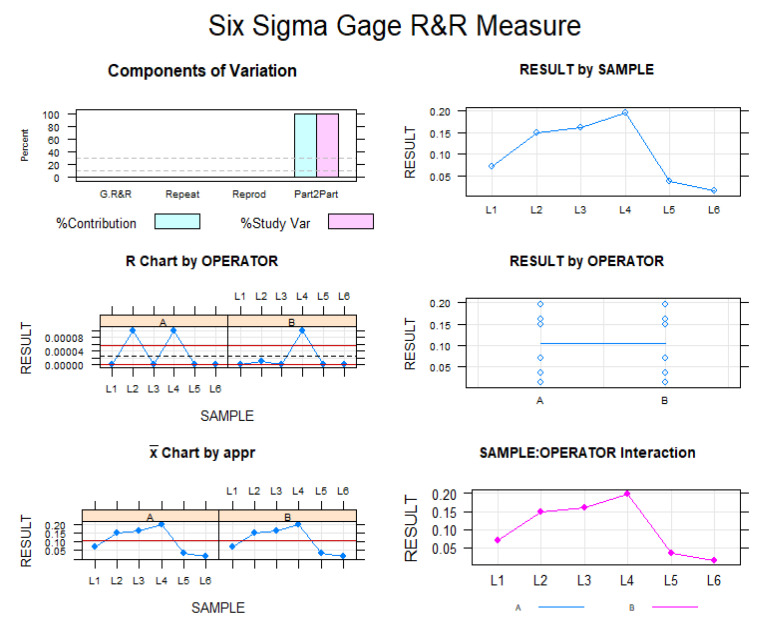
Measurement evaluation chart of the measurements of the volume wear of CM-wire NiTi alloy endodontic reciprocating files after use indicating the difference between the measurements of each observer to evaluate the impact of each variable on the total variation obtained (components of variation) with a mean control chart and a range control chart (R chart by operator and x chart by appr), graphed measurement points (result by sample and result by operator), and interactions (sample–operator interaction). The values are within the confidence limits.

**Table 1 ijerph-19-03203-t001:** Quantitative statistics regarding the wear volume (mm^3^) of the CM-wire NiTi alloy endodontic reciprocating files between the operators.

Operator		*n*	Mean	SD	Minimum	Maximum
A	1	6	0.070	0.000	0.070	0.070
	2	6	0.149	0.000	0.149	0.149
	3	6	0.161	0.000	0.161	0.161
	4	6	0.197	0.000	0.197	0.197
	5	6	0.036	0.000	0.036	0.036
	6	6	0.015	0.000	0.015	0.015
	7	6	0.028	0.000	0.028	0.028
	8	6	0.089	0.000	0.089	0.089
	9	6	0.104	0.000	0.104	0.104
	10	6	0.073	0.000	0.073	0.073
B	1	6	0.070	0.000	0.070	0.070
	2	6	0.149	0.000	0.149	0.149
	3	6	0.161	0.000	0.161	0.161
	4	6	0.197	0.000	0.197	0.197
	5	6	0.036	0.000	0.036	0.036
	6	6	0.028	0.000	0.028	0.028
	7	6	0.089	0.000	0.089	0.089
	8	6	0.104	0.000	0.104	0.104
	9	6	0.073	0.000	0.073	0.073
	10	6	0.028	0.000	0.028	0.028

SD: standard deviation.

## Data Availability

Data are available upon request in accordance with relevant restrictions (e.g., ethical or privacy concerns).

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
