# Peer review of "A Novel Digital Technique to Analyze the Wear of CM-Wire NiTi Alloy Endodontic Reciprocating Files: An In Vitro Study"

_ijerph, 2022, doi:10.3390/ijerph19063203_

Round 1
Reviewer 1 Report
How long did the root canal treatment last in each tooth?
What are the differences between Fig 4B and 4C?
The authors should recognize and discuss the limitations of this method.
Table 1 and 2 could be merged. Correlation coefficients were not reported.
Do Fig 5 and 6 belong in this paper?
To evaluate the wear volume of NiTi endodontic reciprocating files after clinical use.Micro-CT scans were used to evaluate the files preoperatively and postoperatively. The repeatability and reproducibility of this digital measurement technique used to analyze the wear volume was excellent.
• Was the duration of the root canal treatment standardized? • Correlation coefficients were not reported. 5. Are the conclusions consistent with the evidence and arguments presented and do they address the main question posed? Yes.
• What are the differences between Fig 4b and 4C? • The authors should recognize and discuss the limitations of this method. • Table 1 and 2 could be merged. • Do Fig 5 and 6 belong in this paper?
Author Response
Dear Reviewer 1,
I’m pleased to resubmit the manuscript of the work entitled, “Novel Digital Technique to Analyze the Wear of CM‑Wire NiTi Alloy Endodontic Reciprocating Files: An In Vitro Study
Reviewer 1: English language and style are fine/minor spell check required
Response: In order to adapt to the reviewer's 1 comments, we have sent the manuscript to the English Editing Service of MDPI. We attached the Certificate.
Reviewer 1: How long did the root canal treatment last in each tooth?
Response: In order to adapt to the reviewer's 1 comments, we clarify that all root canal systems were instrumented with CM-wire NiTi alloy endodontic reciprocating files (R40, Reciproc; VDW, Munich, Germany) for 73.1 seconds, according to the study performed by Bürklein et al. (Bürklein S, Hinschitza K, Dammaschke T, Schäfer E. Shaping ability and cleaning effectiveness of two single-file systems in severely curved root canals of extracted teeth: Reciproc and WaveOne versus Mtwo and ProTaper. Int Endod J. 2012 May;45(5):449-61. doi: 10.1111/j.1365-2591.2011.01996.x). We have added a sentence in the Material and Methods section.
Reviewer 1: What are the differences between Fig 4B and 4C?
Response: In order to adapt to the reviewer's 1 comments, we clarify that the Figure 4B refers to the isolation and measure of the wear area and the Figure 4C refers to the isolation and measure of the wear volume. We have added a sentence in the Materials and Methods section.
Reviewer 1: The authors should recognize and discuss the limitations of this method.?
Response: In order to adapt to the reviewer's 1 comments, we clarify that the limitations of the novel digital measurement technique are recognized and discussed in the last sentence of the Discussion section: “One limitation observed in the present study is that NiTi alloy endodontic files with smaller diameters, as well as files with metallurgical composition or thermal treatments that allow more flexibility, may compromise the alignment procedure”.
Reviewer 1: Table 1 and 2 could be merged.
Response: In order to adapt to the reviewer's 1 comments, we clarify that there is only one table (Table 1) in the manuscript, there is no Table 2.
Reviewer 1: Do Fig 5 and 6 belong in this paper?
Response: In order to adapt to the reviewer's 1 comments, we clarify that Figures 5 and 6 belong to this study. Both represent the results obtained from the Gage R&R statistical analysis.
Reviewer 1: Correlation coefficients were not reported.
Response: In order to adapt to the reviewer's 1 comments, we clarify that correlation among operators was 1.
We take this opportunity to thank the recommendations and suggestions made by the reviewers to improve the document.
Yours sincerely,
Reviewer 2 Report
In this submission to IJERPH, the authors describe a novel digital measurement technique for analyzing the wear volume of Controlled Memory (CM)–wire NiTi alloy endodontic reciprocating files after clinical use. The authors use ten CM-wire NiTi endodontic reciprocating files in 10 first upper molar teeth within four root canals. The authors subjected the CM-wire NiTi alloy endodontic reciprocating files to a preoperative and postoperative micro-computed tomography (micro-CT) scan to obtain accurate digital imaging and communication. The authors complemented their studies with a repeatability and reproducibility study. The authors conclude that their technique is a repeatable, reproducible, and accurate method of quantifying the wear volume of CM-wire NiTi alloy endodontic reciprocating files after clinical use.
I consider this manuscript to be of interest to endodontic materials researchers as well as readers of this journal. As such, I am relatively supportive of publication with a few minor notes. In particular, there has been much prior work by using polymer-based materials for endodontic as well as flexible wearable devices, which should be mentioned in the next revision:
Materials 2019, 12, 2347
Adv. Mater. 2017, 29, 1605099
In particular, these prior works showed that polymer-based materials can be harnessed for endodontic implementation as well as flexible substrates due to their versatility, which should be mentioned as related work in the next revision of this manuscript. With these minor edits, I would be willing to re-review this manuscript for subsequent publication.
Author Response
Dear Reviewer 2,
I’m pleased to resubmit the manuscript of the work entitled, “Novel Digital Technique to Analyze the Wear of CM‑Wire NiTi Alloy Endodontic Reciprocating Files: An In Vitro Study".
Reviewer 2: I am relatively supportive of publication with a few minor notes. In particular, there has been much prior work by using polymer-based materials for endodontic as well as flexible wearable devices, which should be mentioned in the next revision: Materials 2019, 12, 2347, Adv. Mater. 2017, 29, 1605099. In particular, these prior works showed that polymer-based materials can be harnessed for endodontic implementation as well as flexible substrates due to their versatility, which should be mentioned as related work in the next revision of this manuscript. With these minor edits, I would be willing to re-review this manuscript for subsequent publication.
Response: In order to adapt to the reviewer's 2 comments, we have added these references in the Introduction section.
We take this opportunity to thank the recommendations and suggestions made by the reviewers to improve the document.
Yours sincerely,
Round 2
Reviewer 1 Report
I asked the authors about Fig 5 and 6, since their legends do not belong to the present paper (they are about cement left after debonding via lingual orthodontic treatment). This issue should be addressed.Author Response
Dear Reviewer 1,
I’m pleased to resubmit the manuscript of the work entitled, “Novel Digital Technique to Analyze the Wear of CM Wire NiTi Alloy Endodontic Reciprocating Files: An In Vitro Study
Reviewer 1: I asked the authors about Fig 5 and 6, since their legends do not belong to the present paper (they are about cement left after debonding via lingual orthodontic treatment). This issue should be addressed.
Response: In order to adapt to the reviewer's 1 comments, we apologize to Reviewer 1 comments, because we revised the Figures, but we did not revise the captions. Finally, we have changed the captions of Figures 5 and 6.
We take this opportunity to thank the recommendations and suggestions made by the reviewers to improve the document.
Yours sincerely,